# Cross-synaptic synchrony and transmission of signal and noise across the mouse retina

William N Grimes[1], Mrinalini Hoon[2], Kevin L Briggman[3], Rachel O Wong[2], Fred Rieke[1]*

[1]Department of Physiology and Biophysics, Howard Hughes Medical Institute, University of Washington, Seattle, United States; [2]Department of Biological Structure, University of Washington, Seattle, United States; [3]Circuit Dynamics and Connectivity Unit, National Institute of Neurological Disorders and Stroke, Bethesda, United States

**Abstract** Cross-synaptic synchrony—correlations in transmitter release across output synapses of a single neuron—is a key determinant of how signal and noise traverse neural circuits. The anatomical connectivity between rod bipolar and A17 amacrine cells in the mammalian retina, specifically that neighboring A17s often receive input from many of the same rod bipolar cells, provides a rare technical opportunity to measure cross-synaptic synchrony under physiological conditions. This approach reveals that synchronization of rod bipolar cell synapses is near perfect in the dark and decreases with increasing light level. Strong synaptic synchronization in the dark minimizes intrinsic synaptic noise and allows rod bipolar cells to faithfully transmit upstream signal and noise to downstream neurons. Desynchronization in steady light lowers the sensitivity of the rod bipolar output to upstream voltage fluctuations. This work reveals how cross-synaptic synchrony shapes retinal responses to physiological light inputs and, more generally, signaling in complex neural networks.

*For correspondence: rieke@u.washington.edu

**Competing interests:** The authors declare that no competing interests exist.

**Reviewing editor**: Ronald L Calabrese, Emory University, United States

## Introduction

Everyday activities depend on the reliability of neural computation. Yet the mechanistic building blocks of the underlying circuits can exhibit highly stochastic behavior. Understanding neural computation requires bringing these two perspectives together, in particular, it requires identifying, under physiological conditions, the sources of noise in neural signals and how such noise is controlled (*Deneve et al., 2001*; *Averbeck et al., 2006*). Sensory systems, for example, must encode and transmit physiological stimuli accurately and quickly based on inherently stochastic processes such as transduction and synaptic transmission.

Neural circuits share a number of common architectural features. Cross-synaptic synchrony (CSS)—synchronization of transmitter release at different output synapses of a single neuron—is an important factor in how these common architectural features will impact the coding and transmission of physiological signals (*Salinas and Sejnowski, 2000*; *Rosenbaum et al., 2013*, *2014*). For example, the strength of network correlations generated by divergent synaptic output from a single presynaptic neuron depends directly on CSS. Similarly, if CSS is high, multiple parallel synapses between a pre- and postsynaptic neuron can enhance transmission of upstream noise and mitigate the impact of intrinsic synaptic noise. Circuits using graded signals or action potentials share these issues because the small number of vesicles associated with transmission of physiological signals cause synaptic release to vary even for nominally fixed presynaptic signals such as action potentials (*Del Castillo and Katz, 1954*; *Allen and Stevens, 1994*).

**eLife digest** The human eye is capable of detecting a single photon of starlight. This level of sensitivity is made possible by the high sensitivity of photoreceptors called rods. There are around 120 million rods in the retina, and they support vision in levels of light that are too low to activate the photoreceptors called cones that allow us to see in color. This is why we cannot see colors in the dark.

Signals are relayed through the retina via a circuit made up of multiple types of neurons. The activation of rods leads to activation of cells known as 'rod bipolar cells' which, in turn, activate amacrine cells and ganglion cells, with the latter sending signals via the optic nerve to the brain. All of these neurons communicate with one another at junctions called synapses. Activation of a rod bipolar cell, for example, triggers the release of molecules called neurotransmitters: these molecules bind to and activate receptors on the amacrine cells, enabling the signal to be transmitted.

For the brain to detect that a single photon has struck a rod, the eye must transmit information along this chain of neurons in a way that is highly reliable while adding very little noise to the signal. Grimes et al. have now revealed a key step in how this is achieved.

Electrical recordings from the mouse retina revealed that, in the dark, small fluctuations in the activity of rod bipolar cells lead to the near-deterministic release of neurotransmitters. This reduces the impact of random fluctuations in neurotransmitter release produced at individual synapses and ensures that the signals from rod bipolar cells (and thus from rods) are transmitted faithfully through the circuit with minimal added noise. As light levels increase, this tight synchrony of transmitter release breaks down, reducing the sensitivity to individual photons.

Given that many other brain regions share the features that enable retinal cells to coordinate the release of neurotransmitters, this mechanism might be used throughout the brain to increase the signal-to-noise ratio for the transmission of information through neural circuits.

Two issues have hindered progress in understanding the importance of cross-synaptic synchrony for neural signaling. First, quantitative anatomical information about convergent and divergent wiring patterns is essential for understanding the influence of CSS, but such information is lacking for most brain regions (*Denk et al., 2012*). Second, the experimental conditions for studying neurotransmitter release biophysically often preclude studying physiological signaling in the same circuit. Retinal signaling at low light levels provides an opportunity to tackle these issues directly because of the wealth of available anatomical information (*Kolb, 1970*; *Kolb and Famiglietti, 1974*; *Tsukamoto et al., 2001*; *Tsukamoto and Omi, 2013*) and the ability to study synaptic mechanisms in the context of physiological light stimuli (*Sampath and Rieke, 2004*; *Dunn and Rieke, 2008*; *Oesch and Diamond, 2011*).

In starlight, signals from rod photoreceptors traverse the retina through the specialized rod bipolar circuit (for review see *Bloomfield and Dacheux, 2001*). Behavioral and physiological studies indicate that the sensitivity of this circuit approaches limits set by noise in the rod photoreceptors themselves, indicating that little noise is generated at downstream synapses (for review see *Field et al., 2005*). Rod bipolar cells (RBCs) provide a key component of this circuit; they receive dendritic input exclusively from rods, and provide output to several types of amacrine cells via anatomically distinct and stereotypic connections (*Nelson and Kolb, 1984*; *Tsukamoto and Omi, 2013*). Here, we exploit the distinct anatomical features of these connections to quantitatively characterize the role of cross-synaptic synchrony in retinal signaling at low light levels. We find that RBC output synapses can exhibit near-perfect synchronization for small physiological changes in presynaptic voltage such as those encountered near visual threshold. This reveals a surprisingly low level of stochastic behavior at individual synapses under dark-adapted conditions.

## Results

Our aims were (1) to determine the degree of synchronization of RBC output synapses, (2) to identify the factors that control synchronization, and (3) to measure the impact of synchronization on the transmission of signal and noise through the rod bipolar circuit.

## Same synaptic input, very different signal

AII and A17 amacrine cells provide the two main postsynaptic targets of RBCs. As described below, the connectivity between RBCs and these postsynaptic neurons provides an opportunity (1) to measure cross-synaptic synchrony (CSS) using paired A17 recordings, and (2) to examine its role in neural transmission by recording feedforward signaling in the AII amacrine cell and other downstream circuit elements.

Each rod bipolar cell releases glutamate from ~50 ribbon synapses (*Sterling and Lampson, 1986*; *Tsukamoto et al., 2001*; *Tsukamoto and Omi, 2013*; *Mehta et al., 2014*) (*Figure 1A*). These specialized synapses often exist as dyads, in which each presynaptic ribbon is shared by two postsynaptic targets, most commonly one AII (*Figure 1A*, purple) and one A17 (*Figure 1A*, green) amacrine cell. AII and A17 amacrine cells differ in morphology, wiring configuration and functional contributions to retinal processing. AII amacrine cells convey rod-mediated signals to ganglion cells via gap junctions made with the axon terminals of On cone bipolar cells and via glycinergic synapses made with both the axon terminals of Off cone bipolar cells and the dendrites of Off ganglion cells. They have a compact dendritic field (~40 μm) and receive direct synaptic input from ~10 RBCs. A17 amacrine cells provide GABAergic feedback inhibition to RBC axon terminals. They have a wide dendritic field (~300 μm) and collect input from more than 100 RBCs, which provide their only known source of excitation. Central to our work here, each RBC can make tens of synaptic contacts onto a single AII amacrine cell (*Tsukamoto and Omi, 2013*), but typically only one synaptic contact onto a single A17 (*Ellias and Stevens, 1980*; *Nelson and Kolb, 1985*; *Zhang et al., 2002*; *Grimes et al., 2010*).

Spontaneous excitatory synaptic inputs to voltage-clamped AII (*Figure 1B*) and A17 (*Figure 1C*) amacrine cells differed dramatically (identical recording conditions, see 'Materials and methods'). Both cell types received substantial excitatory input in complete darkness as evinced by the suppression of holding current and noise by 10 μM NBQX, an AMPA receptor antagonist (data not shown); however, spontaneous current fluctuations observed in AII amacrines were much larger in amplitude ($\sigma^2 = 1000 \pm 154$ pA$^2$, n = 20) than those observed in A17 amacrines ($\sigma^2 = 72 \pm 22$ pA$^2$, n = 17; *Figure 1E*).

AII amacrine cells receive conventional excitatory (i.e., glutamatergic) synaptic input from RBCs and direct electrical input from other AIIs and cone bipolar cells via gap junctions. Deletion of connexin-36 disrupts the gap junctional input (*Gjd2* knockout mouse, note: this mouse is also commonly referred to as the Cx36 knockout mouse; *Deans et al., 2001*, *2002*); under these conditions, AII input currents continued to exhibit large variability in darkness (*Figure 1D,E*). Variability in the AII inputs was insensitive to pharmacological block of the receptors mediating feedforward (to the AII) and feedback (to the RBC axon terminal) inhibition (*Figure 1E*). Together, these results indicate that the large spontaneous fluctuations in the AII inputs arise from excitatory RBC inputs and do not require synaptic inhibition.

How can synaptic inputs from RBCs to AII and A17 amacrine cells differ so markedly? Multiple factors, such as differences in the cells' electrical properties, could contribute; we hypothesized that a key factor was differences in the connectivity of AII and A17 amacrine cells with RBCs and synchronization of output synapses within individual RBC axon terminals. Since A17 amacrine cells receive input, on average, from one ribbon-type synapse per RBC, their synaptic input should not be affected by synchronization across ribbons. AII amacrine cells, however, receive inputs from multiple synapses per RBC and hence their inputs should be shaped by RBC CSS. In support of this synchronization hypothesis, closer examination of the noisy AII input currents in *Gjd2* knockout mice revealed that the largest spontaneous current fluctuations were many times larger in amplitude and total charge than the average miniature excitatory postsynaptic current (mEPSC; see 'Materials and methods'; *Figure 1D*).

The differences in RBC connectivity with AII and A17 amacrines and in excitatory inputs to the two postsynaptic amacrine cells (in darkness) suggest that CSS substantially shapes RBC synaptic output. The RBC's CSS cannot be measured under dark-adapted conditions using imaging approaches because even two-photon (i.e., infrared) imaging produces too much rod activation to maintain the retina in a dark-adapted state (*Euler et al., 2009*). Instead, as described below, we took advantage of the sparse, stereotyped connectivity between RBCs and A17s to characterize the CSS of RBCs under physiological conditions.

## Synchrony of RBC output

Each A17 amacrine cell is contacted by a large fraction of the RBCs within its dendritic field (~50% in rabbit, *Zhang et al., 2002*); therefore, pairs of A17s with highly overlapping dendrites receive synaptic contact from many of the same RBCs (i.e., RBCs that are common to both A17s; *Figure 2A,B*). Because

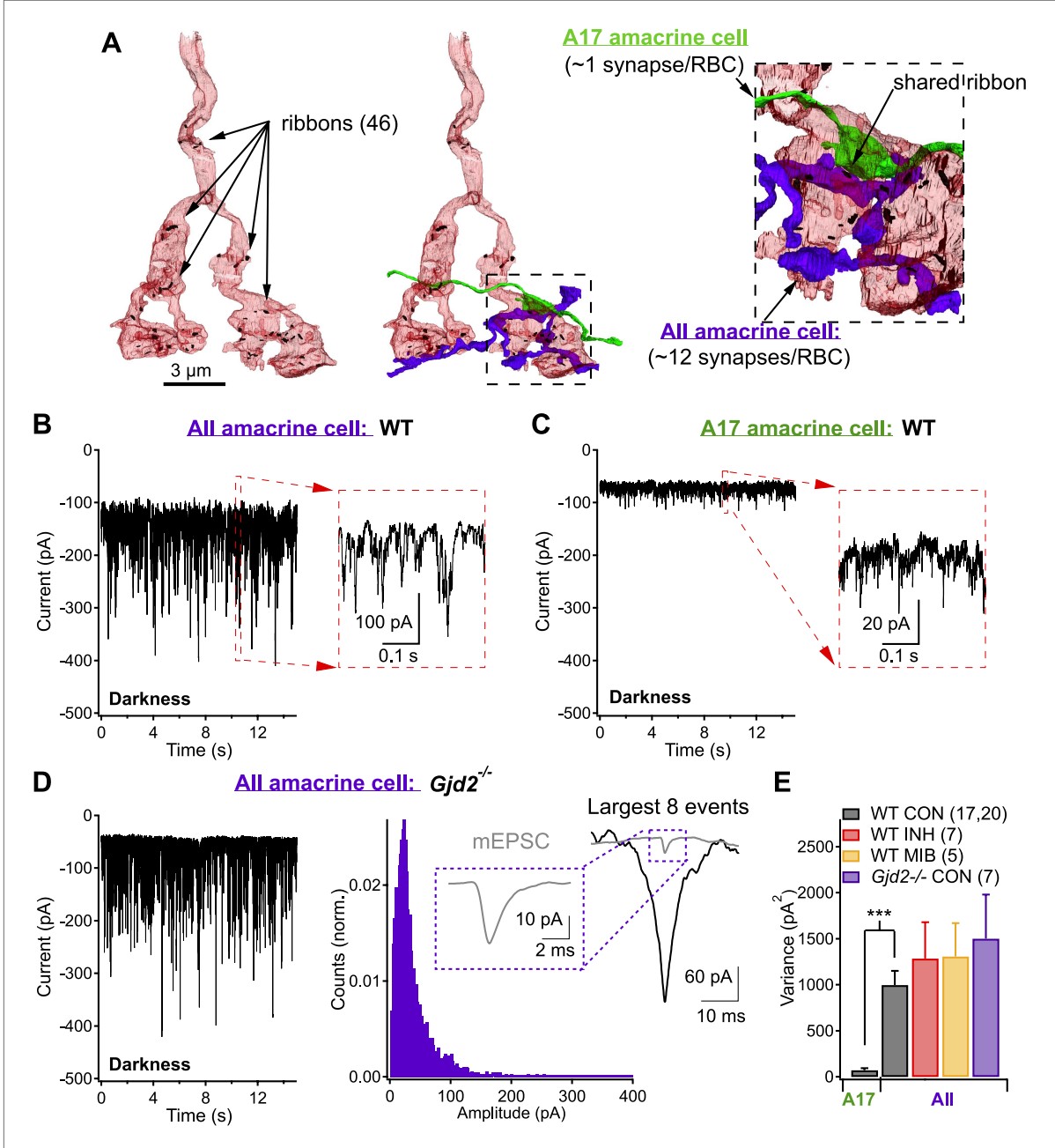

**Figure 1**. Same presynaptic neuron, very different postsynaptic noise properties. (**A**) Three-dimensional EM reconstruction of a rod bipolar cell axon terminal (pink), presynaptic ribbons are represented with black markers (46 in total). AII (purple) and A17 (green) amacrine cells are complimentary postsynaptic partners at individual RBC ribbon synapses, but unlike the A17, AIIs receive synaptic input from multiple presynaptic ribbons. (**B** and **C**) Voltage-clamp recordings from AII (**B**) and A17 (**C**) amacrine cells ($V_{hold}$ ~−60 mV) in retinas from wild-type mice demonstrate that tonic excitatory synaptic input (from RBCs) to these two cell types can be very different under physiological recording conditions. Under dark-adapted conditions large noise events are observed at RBC→AII connections (**B**) but not at RBC→A17 connections (**C**). (**D**) AII recordings from retinal slices lacking Cx36-containing gap junctions (i.e., *Gjd2* knockout mouse, where electrical synapses between AII amacrine cell dendrites and On cone bipolar axon terminals have been eliminated) exhibited similar behavior to recordings from wild-type retina. Under these conditions synaptic events were analyzed. Inset: fast synaptic events (with 10–90% rise time ≤1 ms, i.e., miniature excitatory postsynaptic current or mEPSC) exhibited amplitudes that were less than a tenth of that of the largest events. (**E**) Population statistics for the synaptic noise recorded from AII and A17 amacrine cells in wild type and $Gjd2^{-/-}$ recordings under dark-adapted conditions (control or drugs). On average, noise recorded from AII amacrine cells (WT) was >10 times larger than noise recorded from A17 amacrine cells (unpaired *t* test p = 3 × 10⁻⁶). Neither an inhibitory cocktail (2 μM Strychnine, 20 μM SR95531 and 50 μM TPMPA) or mibefridil (10 μM, T-type Ca$_v$ channel antagonist) caused a significant change in the noise recorded from AII amacrine cells in darkness.

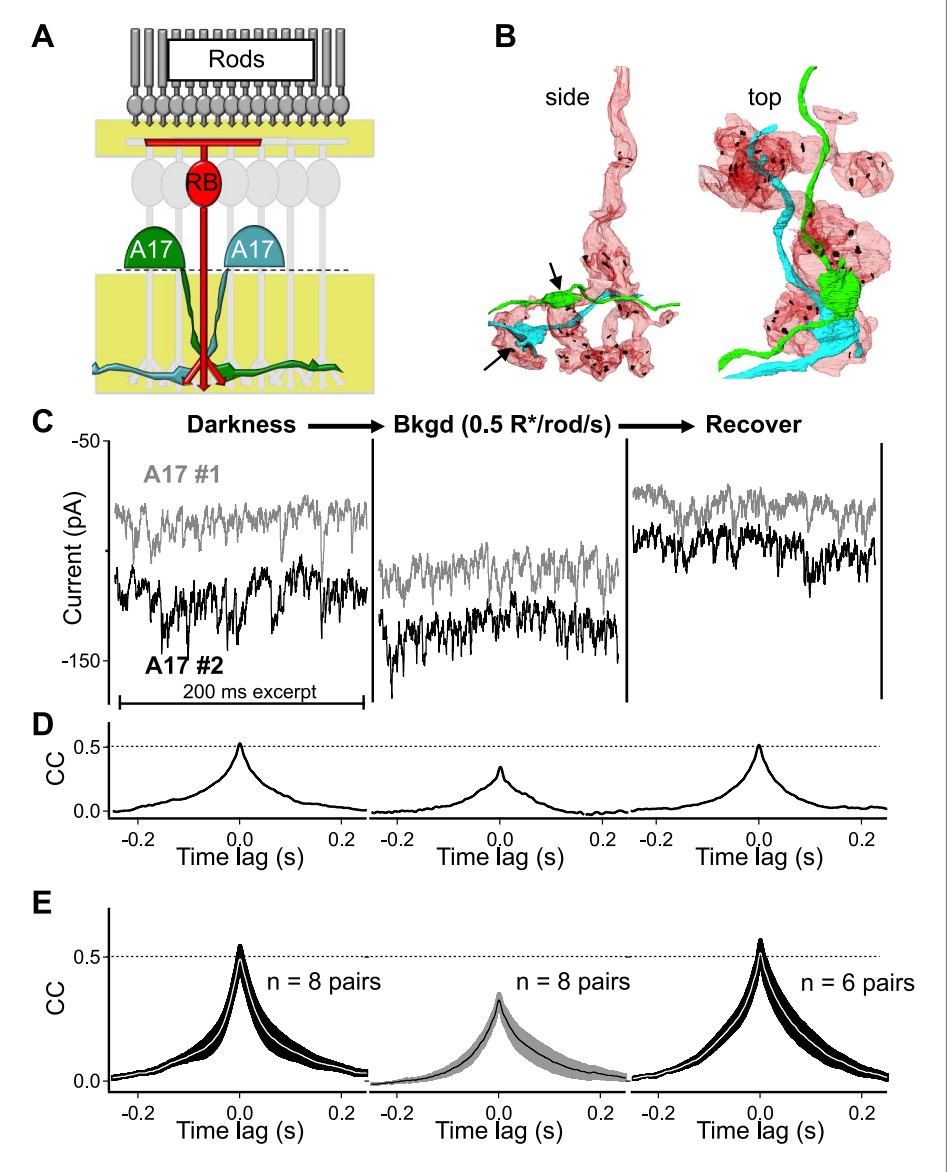

**Figure 2**. Strong covariation in overlapping A17 amacrine cells reflects highly synchronized cross-synaptic release from individual RBCs under dark-adapted conditions. (**A**) Paired recordings from neighboring A17 amacrine cells in the wild type retinal slice preparation were used to measure the CSS of RBC output under near-physiological conditions. (**B**) Pairs of highly overlapping A17 amacrine cells contact many of the same RBCs but at different synaptic locations (arrows). Same RBC serial EM reconstruction as in **Figure 1** but with an additional reconstructed A17 amacrine cell dendrite from a different A17 (AII is removed). (**C–E**) Paired recordings from neighboring A17 amacrine cells revealed strong covariation in excitatory synaptic input from RBCs under dark-adapted conditions. Dim backgrounds increased presynaptic release (**C**) but decreased correlated activity in neighboring A17 amacrine cells (**D** and **E**; p = 0.0053 for change relative to dark, n = 8 pairs). Upon returning to darkness for ~2 min the strong covariation of presynaptic input recovers to that observed before the background was presented. (**E**) Population data for cross-correlation functions in darkness (left), 0.5 R*/rod/s (middle) and after returning to darkness (i.e. recover, right). Thick lines represent the mean, shaded regions represent ±SEM.

single ribbons typically provide input to an AII–A17 dyad and a single RBC typically contacts an A17 only once (**Figure 1A**), highly overlapping A17s receive input from different ribbon-type synapses made by many of the same (i.e., common) RBCs (**Figure 2B**). Thus synchronized output from synapses within individual RBCs should cause the synaptic input to nearby A17 amacrine cells to covary.

Paired recordings from neighboring A17 amacrine cells (distance between somas < 80 µm) revealed strong correlations in excitatory synaptic input in the dark (The peak of the cross-correlation function in darkness, that is, the Dark $CC_{peak}$, was 0.51 ± 0.03, n = 8 pairs; *Figure 2C–E*). Eliminating excitatory synaptic transmission between RBCs and A17s with the AMPA-receptor antagonist NBQX eliminated the correlations (data not shown) and dim backgrounds reversibly reduced correlation strength (dim $CC_{peak}$ = 0.34 ± 0.02, p = 0.0053, n = 8 pairs; *Figure 2C–E*). These results are consistent with strong synchronization of RBC synapses, but they could also reflect electrical coupling between A17 amacrines and/or divergence of upstream rod noise to two or more RBCs.

Direct measurements of electrical coupling between highly overlapping A17s revealed an electrical resistance of 9.8 ± 1.3 GΩ (n = 6 pairs; *Figure 3A,B*), more than 30 times the average A17 input resistance (~300 MΩ). Substantial dark correlations were present in pairs with the highest resistance (>15 GΩ), suggesting at most a small contribution from electrical coupling. We did not attempt to eliminate electrical coupling using genetic manipulations because the connexins forming gap junctions between A17s have not been identified. However, pharmacological experiments described below (see section 'Redundant connections and CSS scale dark noise transmission') provide additional evidence for little contribution of gap junctions to correlations in A17 signals. Contributions from upstream divergence were also minimal, as revealed by direct recordings from pairs of RBCs with touching somas (*Figure 3C,D*). Spontaneous excitatory synaptic input to neighboring RBCs was at most weakly correlated in darkness (0.03 ± 0.02, n = 6; *Figure 3E*), indicating minimal correlations in the signals of neighboring RBCs due to rod divergence. Together these experiments indicate that synchronized release from different synapses made by the same RBC dominates the measured correlations in inputs to nearby A17 amacrine cells.

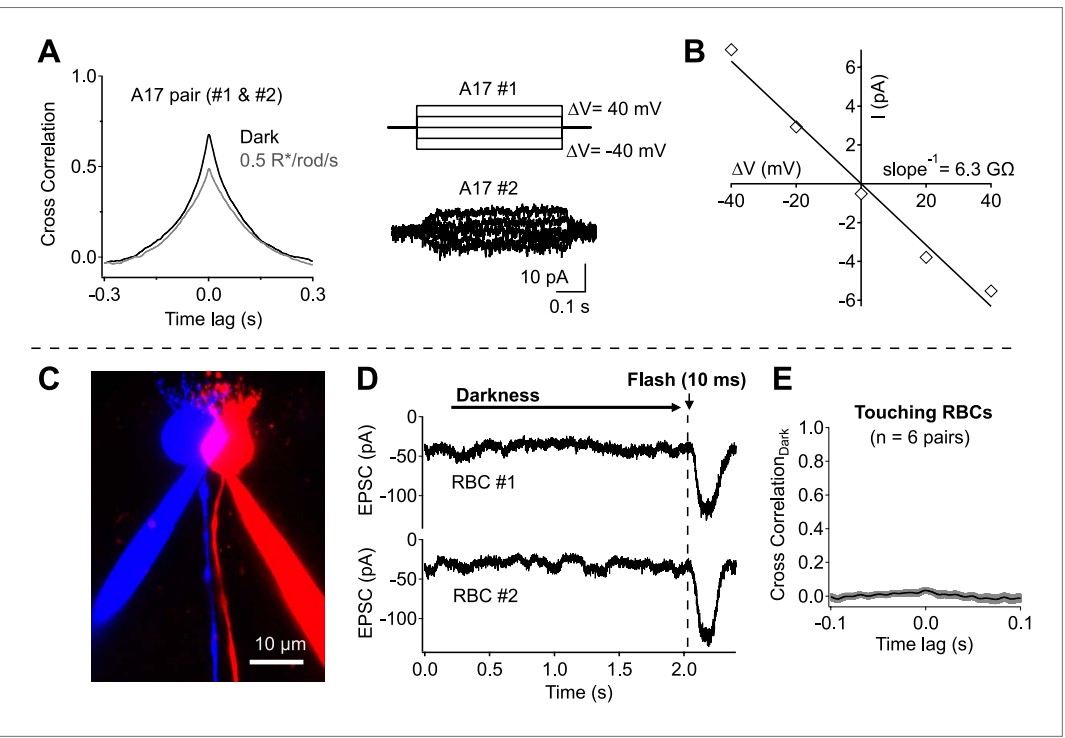

**Figure 3**. Network divergence and electrical coupling only weakly contribute to correlations observed in highly overlapping A17 amacrine cells. (**A** and **B**) Overlapping A17s exhibit weak electrical coupling. (**A**) Example recording: strong correlations are observed in overlapping A17 amacrine cells (left), even in the absence of significant electrical coupling (right). (**B**) The gap junctional resistance was estimated by determining the slope of the ΔV-I relationship. (**C–E**) Dendritic input to neighboring RBCs is only weakly correlated in darkness. (**C**) Confocal reconstruction of a paired recording from RBCs with touching somas. (**D**) Example traces from touching RBC pair. Each recording trial consisted of 2 s of complete darkness followed by a 10 ms flash (to monitor sensitivity). (**E**) Cross correlations were derived for each recording pair before averaging across cells (mean ± SEM). These experiments were conducted using wild-type retinal slices.

The strength of correlations in spontaneous inputs to neighboring A17 amacrine cells in darkness indicated a surprising level of synaptic synchronization considering the lack of visual stimuli and the fact that the presynaptic RBCs are non-spiking neurons. Thus even if the two recorded A17s receive input from an identical set of RBCs, correlation strengths near 0.5 require that two synapses made by the same RBC must be coactive at least half the time. Thus strong synaptic synchronization requires large, coordinated increases in the probability of release (a notion supported by the electrically compact nature of the RBC, *Protti and Llano, 1998*) and low intrinsic variability at individual synaptic connections (see 'Discussion'). The remaining experiments investigate the origin and functional impact of such coordinated release.

## Quantifying cross-synaptic synchrony

The strength of correlations in the spontaneous inputs to neighboring A17s will be controlled by the extent to which the cells receive input from common RBCs and by the strength of CSS in the RBC output. As described below, quantitative anatomical measurements of RBC-A17 connectivity (*Figure 4*) allowed us to relate the measured input correlations to nearby A17 amacrines to the strength of CSS in RBC output. This analysis indicates near-perfect synchronization in the output of RBC synapses in the dark.

Assuming that the measured correlations illustrated in *Figure 2* reflect entirely synchronization of RBC synapses, the measured correlation strength (i.e., peak of the cross-correlation function) can be expressed in terms of the RBC's CSS ($\beta_{sync}$), the fraction of RBCs that are common to both A17s (i.e., $P_{shared}$; see 'Materials and methods'), the density of RBC connections, and an electrotonic scaling ($\gamma$) accounting for the attenuation of distal inputs when measured at the soma. Assuming complete dendritic overlap of two neighboring A17 amacrine cells, this relationship can be approximated by summing across circular rings centered on the soma:

$$CC_{t=0} = \frac{\beta_{sync} \sum_r P_{shared}(r) n_r \exp(-2(r + r_0)/\gamma)}{\sum_r n_r \exp(-2(r + r_0)/\gamma)}$$

(1)

where r is the mean radial distance from the soma for a given ring (*Figure 4A,D*), $r_0$ is the length of the initial descending dendrites (see 'Materials and methods'), and $n_r$ is the number of RBCs contacted within each concentric ring (*Figure 4F*). The numerator in *Equation 1* represents the fraction of inputs to one A17 shared with the other, weighted by the electrotonic attenuation and the degree of synchrony. The denominator represents the total input to the cells. The assumptions in the model—for example, complete dendritic overlap of the two cells—cause *Equation 1* to underestimate the synchrony required to explain a given correlation strength.

We estimated the anatomical parameters of *Equation 1* by filling individual A17 amacrine cells with Lucifer yellow (*Figure 4A*) and labeling RBCs using antibodies against PKCα (*Haverkamp and Wassle, 2000*) (see 'Materials and methods'). A17 synaptic inputs were located by the small, bead-like varicosities along their dendrites (*Figure 4A*). Synaptic contacts between RBCs and A17s were identified by assessing volume overlap in 3-D between the postsynaptic A17 varicosities and the presynaptic PKCα-labeled RBCs (*Figure 4B,C*; see 'Materials and methods'). Connectivity was assessed in concentric rings (Δr = 20 μm) centered on the cell body of each A17 and plotted as a function of radial distance (*Figure 4E*). $P_{shared}$ in *Equation 1* represents the fraction of the RBC inputs to a given A17 that are shared by an overlapping A17; assuming that the two A17s are independently wired, $P_{shared}$ is equivalent to the A17→RBC connection probability (*Figure 4E*). *Figure 4F* plots the number of synaptic contacts made onto a single A17 per ring ($n_r$). The electronic distance (γ = 32.3 μm) for the A17 dendrites was taken from previous work (*Grimes et al., 2010*).

Given the parameters measured above, *Equation 1* provides a correspondence between the measured strength of correlations in the inputs to neighboring A17s and the strength of CSS across RBC synapses (purple line in *Figure 4G*). In other words, *Equation 1* allows us to map correlation strength (x-axis in *Figure 4G*) to CSS strength (y-axis). The slice preparations used for the paired A17 recordings will disrupt the distal dendrites. To correct for this, we reduced $n_r$ for rings with a radius exceeding 20 μm (see 'Materials and methods'); this correction affected the relation between CSS and correlation strength less than 5%. Because our focus was on using A17s to monitor RBC output, our results are otherwise relatively insensitive to a loss of dendrites in slicing.

Applying *Equation 1* to the A17 paired recordings, we estimate that the measured peak cross-correlation of 0.51 ± 0.03 corresponds to a RBC CSS value of 0.80 ± 0.08 in darkness. Dim backgrounds

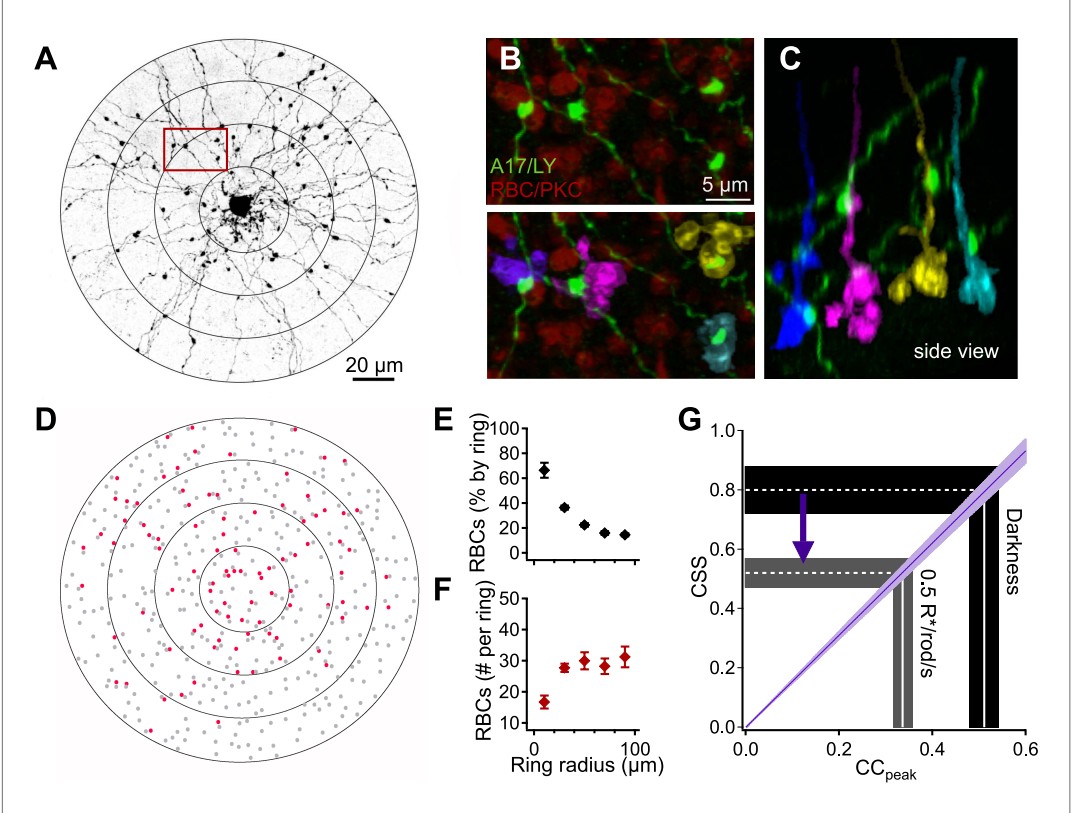

**Figure 4**. Interpreting A17-A17 correlations in terms of RBC cross-synaptic synchrony. A17 amacrine cell-RBC connectivity was assessed using immunohistochemistry and single-cell injections. (**A**) A17 amacrine cells have long, thin neurites that are studded by synaptic varicosities. Connectivity between RBCs and Lucifer-yellow (LY) injected A17s were determined within 20 µm concentric rings centered on the soma of the injected A17 cell. The outermost ring from panels **A** and **D** was removed from the image for better viewing of the proximal dendrites but were included in all analyses. (**B**) Inset from **A**, RBCs were labeled using antibodies against PKCα (red). Synapses between the RBC and A17 cells were identified by sites of appositions between A17 varicosities and the RBC axon terminal (see 'Materials and methods'). Axonal boutons of four RBCs are colorized separately. (**C**) Orthogonal rotation of the image stack showing a side view of the four RBC axon terminals connected to the A17 amacrine cell in **A** and **B**. (**D**) Connectivity map for the A17 cell example in **A**. Red dots represent the axons of connected RBCs, and gray dots represent the axons of RBCs that did not contact the A17. The percentage (**E**) and number (**F**) of RBC connections was determined as a function of radial distance for four injected A17s from four different animals. A17 dendrites traverse ~40 µm of the inner plexiform before reaching sublaminas 4 and 5 (where they make the majority of their synaptic contacts with RBCs), therefore, the most central concentric ring actually corresponds to dendritic distances between 40 and 60 µm, the second ring corresponds to dendritic distances between 60 and 80 µm and so on. (**G**) Changes in the peak amplitude of the cross correlation function reflect luminance-dependent changes in cross-synaptic synchronization as determined by the connectivity and *Equation 1* (purple line). Data are presented as mean ± SEM; SEMs are represented by error bars (**E** and **F**) or shaded regions (**G**). These anatomical experiments were conducted on whole mount retinas taken from *Igfbp2*-GFP mice. Also see *Figure 4—figure supplement 1*.

The following figure supplement is available for figure 4:

**Figure supplement 1**. Images from *Igfbp2*-GFP C57BL6 retina.

reversibly reduced the synchrony of RBC output (CSS: 0.52 ± 0.05, n = 8 pairs; *Figure 4G*). This brings us to two key results. (1) In the dark, release from RBC ribbons is nearly perfectly synchronized. This high CSS, together with differences in connectivity, can at least partially account for the dramatic differences in the properties of RBC synaptic inputs to AII and A17 amacrine cells in darkness (*Figure 1*). (2) This synchronization decreases with steady light.

## Redundant connections and CSS scale dark noise transmission

What produces a high degree of CSS in the RBC's output in the absence of visual stimuli? First, synaptic failures and other sources of variability at individual synapses must be minimized; a full complement of releasable vesicles and multi-vesicular release likely contribute to minimizing intrinsic synaptic variability at individual RBC synapses (see 'Discussion'). Second, different RBC synapses must experience common fluctuations in release probability so that they become coactive. Fluctuations in the dendritic synaptic input to RBCs from rod photoreceptors and consequent fluctuations in RBC voltage could cause release probability to covary. The pharmacological experiments described below support this proposal.

Rod photoreceptors provide input to RBC dendrites at sign-inverting glutamatergic synapses containing mGluR6 postsynaptic receptors. To reveal the role of upstream rod noise in controlling RBC output, we used agonists and antagonists of these receptors while recording from AII and A17 amacrine cells (*Figure 5*). After collecting dark records from an AII or A17 amacrine cell, RBC dendritic input was suppressed by the mGluR6 agonist APB (*Slaughter and Miller, 1981*). APB increases mGluR6 activity and produces a clear reduction in the mean RBC synaptic input (*Sampath and Rieke, 2004*). APB substantially decreased both the holding current and noise observed in the inputs to AII and A17 amacrine cells, indicating that release from RBCs had been strongly suppressed (*Figure 5A–C*).

Next, we added the mGluR6 antagonist LY341495. A mixture of receptor agonist and antagonist should suppress RBC voltage fluctuations while permitting control of the mean RBC voltage and release rate via changes in the agonist/antagonist ratio (*Ala-Laurila et al., 2011*). Indeed, antagonist concentrations could be identified that produced postsynaptic holding currents, and hence RBC synaptic release rates, near those in darkness (dashed lines in *Figure 5A,B*). Similar agonist/antagonist mixtures maintained the mean RBC voltage while suppressing both light responses and dark noise (*Figure 5—figure supplement 1*). This manipulation, however, did not restore noise in the inputs to AII and A17 amacrine cells to its dark level; the suppression of noise was particularly clear in the AII inputs (*Figure 5C*). Thus similar release rates (dark vs appropriate agonist/antagonist mixture) can produce very different levels of synaptic noise. The larger change in variance of the AII inputs compared to the A17 inputs is consistent with the differences in connectivity and a role of CSS in causing small changes in RBC voltage to produce large changes in AII input (*Figure 1*).

The sensitivity of synaptic noise to suppressing fluctuations in RBC dendritic input suggests that RBC voltage fluctuations synchronize synaptic release in the dark. If this is the case, then the strength of correlated synaptic input to neighboring A17s (i.e., CSS) should be sensitive to suppressing RBC voltage fluctuations. Thus, we repeated the pharmacological manipulations of RBC dendritic inputs while recording from A17 pairs (*Figure 5D*). In control conditions, the peak cross-correlation for these pairs was $0.43 \pm 0.05$ ($CSS_{dark} = 0.71 \pm 0.08$, n = 4 pairs). Suppressing RBC synaptic release with APB eliminated correlations ($4 \pm 2\%$ of that in darkness). Restoring the mean release rate with the mGluR6 antagonist modestly increased correlation strength (compared to APB), but it remained significantly less than that in the dark (peak correlation $0.11 \pm 0.03$, corresponding to an estimated CSS of $0.18 \pm 0.05$, p = 0.0034, *Figure 5D,E*).

The agonist/antagonist experiments also provide additional evidence against a primary role for gap junctions between A17s in correlating responses of nearby cells. The cross-correlation measures the fraction of the total noise in one cell that is correlated with noise in another cell. Non-rectified electrical connections between two A17s should cause a fraction of the electrical signal of one A17 to be shared with the other. Correlations in the A17 signals produced by gap junctions should hence be insensitive to decreasing the total noise in the A17 signals by suppressing fluctuations in RBC input as the same fraction of noise should remain correlated. The sensitivity of correlation strength to decreasing RBC voltage fluctuations pharmacologically indicates that gap junctions play at most a modest role in producing correlations, confirming the importance of synchrony across RBC synapses.

These results indicate that CSS strength depends on whether noise in the RBC output is produced by upstream sources or is intrinsic to the synapse (*Figure 5*). Specifically, high CSS in the dark is generated by fluctuations in RBC dendritic input and the resulting fluctuations in RBC voltage.

## Light-dependent changes in CSS and noise transmission

Gain is high within the rod bipolar pathway in the dark (*Dunn et al., 2006*), helping ensure that signal and noise inherited from the rod photoreceptors rather than noise introduced later in retinal processing

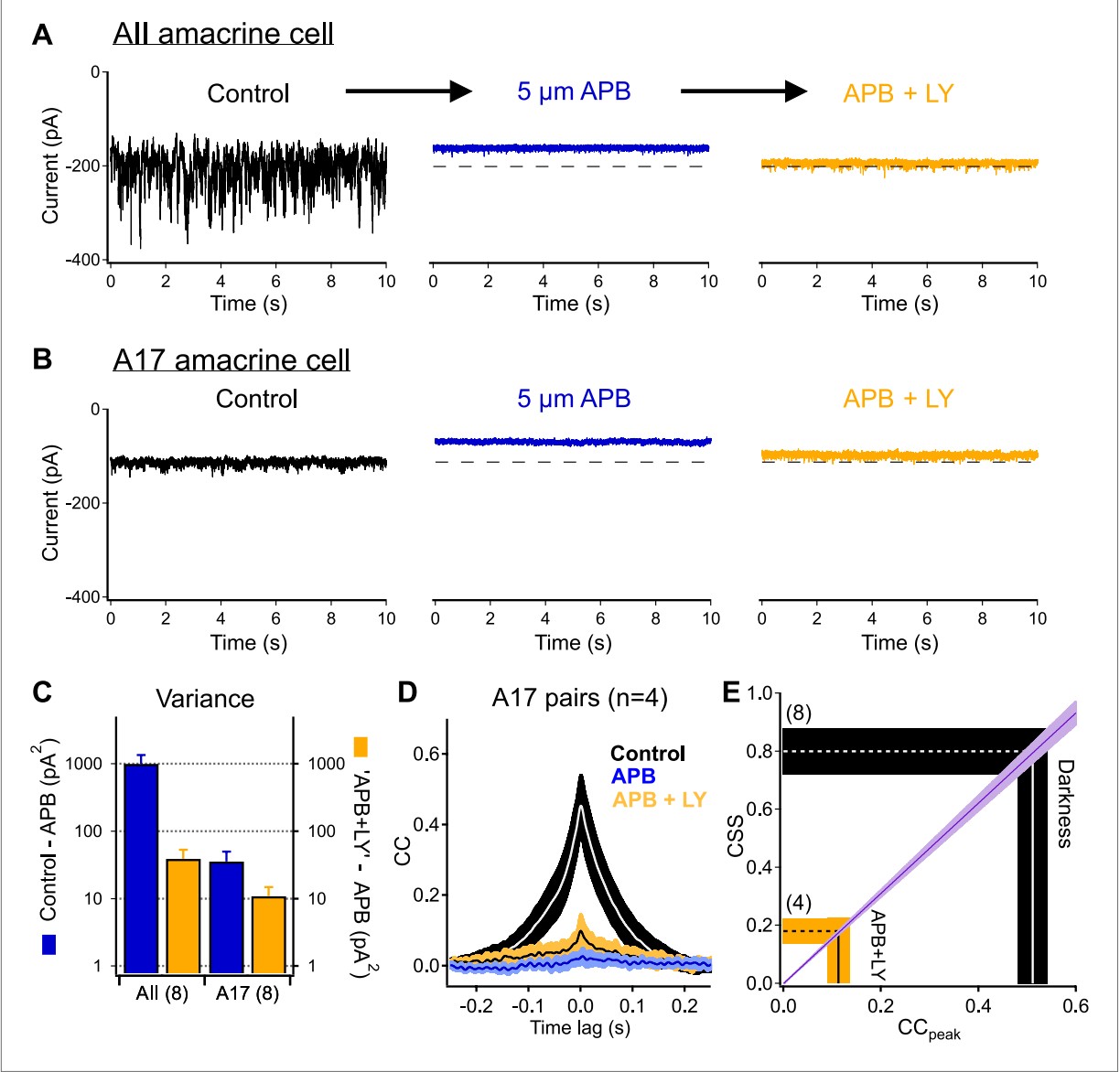

**Figure 5**. Highly synchronized synaptic noise at RBC→AII connections under dark-adapted conditions is driven by upstream rod-dependent noise. (**A**–**B**) mGluR6 agonists and antagonists can be used to override rod→RBC synaptic connections and probe cross-synaptic release properties at RBC→AII connections. Application of the mGluR6 agonist APB (5 μM, *blue-middle*) hyperpolarizes the RBCs and shuts down synaptic transmission (i.e., output) to the postsynaptic AII (**A**) and A17 (**B**) amacrine cells. Addition of the mGluR6 antagonist LY (0.5–2 μM, *yellow-right*) restores tonic release from RBC output synapses (i.e., similar holding current), however, RBC output synapses are now insensitive to fluctuations in transmitter release between rods and RBCs. (**C**) Summary graph comparing network noise properties observed by AII and A17 amacrine cells (n = 8 for all bars). The AII amacrine cell inherits (from RBCs) an order of magnitude more network noise than the A17 amacrine cell (*blue*: $\sigma^2_{Con} - \sigma^2_{APB}$) but recovers only a small fraction of this noise when tonic release from RBC synapses is restored (*yellow*: $\sigma^2_{LY+APB} - \sigma^2_{APB}$). (**D**) Bath application of 'APB' and 'APB + LY' strongly suppress correlated activity in overlapping A17s (APB: p = 0.0088 for change relative to dark; APB + LY: p = 0.0034 for change relative to dark; n = 4 pairs). (**E**) Although the majority of tonic presynaptic glutamate release can be recovered in the presence of 'APB + LY', CSS measurements indicate that RBC output synapses are highly desynchronized under these conditions, thus partially explaining the differences in recovered variance in the AII and A17 amacrine cells. Also see *Figure 5—figure supplement 1*.

The following figure supplement is available for figure 5:

**Figure supplement 1**. Bath application of a solution containing 1 μM LY341495 and 5 μM APB suppresses dendritic input and voltage fluctuations in RBCs.

dominates activity in downstream cells (e.g., AII amacrine cells). Redundant connections between RBCs and AII amacrine cells and high CSS at these connections provide key elements of such amplification. With increasing light level, the rod bipolar pathway ceases to be the sole route for signals to traverse the retina, as rod and cone signals are conveyed to ganglion cells through the cone bipolar circuits (*Xin and Bloomfield, 1999*; *Deans et al., 2002*; *Trexler et al., 2005*; *Manookin et al., 2008*). Under these conditions, the need for a high gain pathway is supplanted by a need to suppress noise so as not to contaminate signals in the cone bipolar circuits. Indeed, as described below, we find that noise transmission and CSS decrease with increasing light level.

Transmission of noise from RBCs to AII amacrine cells was assessed over a 1000-fold range of light levels (~0.5-500 R*/rod/s; *Figure 6*). RBCs depolarize by ~10 mV over this light range, while the mean excitatory synaptic input to AII amacrines decreases (*Jarsky et al., 2011*; *Grimes et al., 2014*). The variance of the RBC signals changed by less than a factor of two across this luminance range (*Figure 6A,D*), while the variance of the AII amacrine signals decreased 10-fold (*Ke et al., 2014*) (*Figure 6B,D*). AII noise also decreased in retinas lacking gap junctions between AII dendrites and On cone bipolar axons (i.e., Gjd2 knockout mouse; *Figure 6D*), indicating it was a property of the RBC input to the AII.

Steady light changed the kinetics of the noise in the AII inputs more than the kinetics of noise in the RBC voltage, as determined from autocorrelograms (*Figure 6E*). Specifically, the broad temporal correlations characteristic of the AII input currents at 0.5 R*/rod/s or in the dark were largely absent at 500 R*/rod/s, while the kinetics of the voltage fluctuations in presynaptic RBCs changed relatively little (*Figure 6E*). The rapid kinetics of noise in the AII inputs at 500 R*/rod/s matched the kinetics of noise in the presence of the mGluR6 agonist/antagonist mixture introduced in *Figure 5* (FWHM of the autocorrelation function was 3.7 ± 0.3 ms in steady light vs 4.3 ± 0.3 ms in drugs, light n = 5, drugs n = 4; *Figure 6F*). The broad temporal correlations in the dark are consistent with synchronized release occurring with temporal correlations dictated by the kinetics of RBC voltage fluctuations. The narrow correlations are consistent with release occurring independently of RBC voltage fluctuations. This suggests that the decrease in correlation width produced by the agonist/antagonist mixture or by steady light reflects a relative increase in asynchronous release.

We recorded from A17 pairs to test for a change in CSS with increasing light. Steady light producing 500 R*/rod/s reduced the peak correlations by more than ~50% (*Figure 6G*) and reduced the estimated CSS to 0.36 ± 0.08 (*Figure 6H*). Rod input to the RBC dendrites continued to produce some correlated output even in bright steady light, as indicated by comparison to the lower correlations (and CSS) observed when using the mGluR6 agonist/antagonist mixture (*Figure 6H*).

The experiments in this section indicate that CSS of the RBC output synapses decreases with increasing light level, such that the synapses becomes less sensitive to fluctuations in RBC voltage. This result is consistent with recent studies showing that the increase in mean RBC voltage with increasing light level produces presynaptic depression by reducing the pool of available vesicles at the RBC→AII synapse (*Dunn and Rieke, 2008*; *Jarsky et al., 2011*; *Oesch and Diamond, 2011*). Differences in vesicle availability across synapses will increase their variability and thus reduce CSS. Lowered CSS in turn causes less effective transmission of noise or small signals produced upstream in the rod photoreceptors.

## Synchronized dark noise in the RB pathway drives correlated activity in retinal ganglion cells

What impact does the RBC's CSS have on downstream signaling in the retinal output neurons (i.e., retinal ganglion cells, or RGCs)? To answer this question, we recorded synaptic input to pairs of On alpha and Off sustained RGCs (*Figure 7*). At low light levels, excitatory inputs to On alpha RGCs originate from modulation of the On cone bipolar synaptic output via electrical coupling with AII amacrine cells, while inhibitory input to Off sustained RGCs originates directly from glycinergic output of the AII (*Murphy and Rieke, 2008*). Thus the primary source of correlations in these signals comes from fluctuations in AII voltage, which, as we show above, is sensitive to CSS.

Excitatory input to the On alpha RGC and inhibitory input to the Off sustained RGC were highly correlated in the dark ($CC_{peak}$ = 0.43 ± 0.04, n = 5 pairs; *Figure 7B,C*). Dim steady light (0.5 R*/rod/s) increased tonic synaptic input to the two cells but, if anything, decreased correlations in synaptic inputs ($CC_{peak}$ = 0.38 ± 0.04, n = 5 pairs; *Figure 7C*). Bath application of the mGluR6 agonist APB suppressed synaptic input and eliminated correlated activity in On alpha-Off sustained RGC pairs

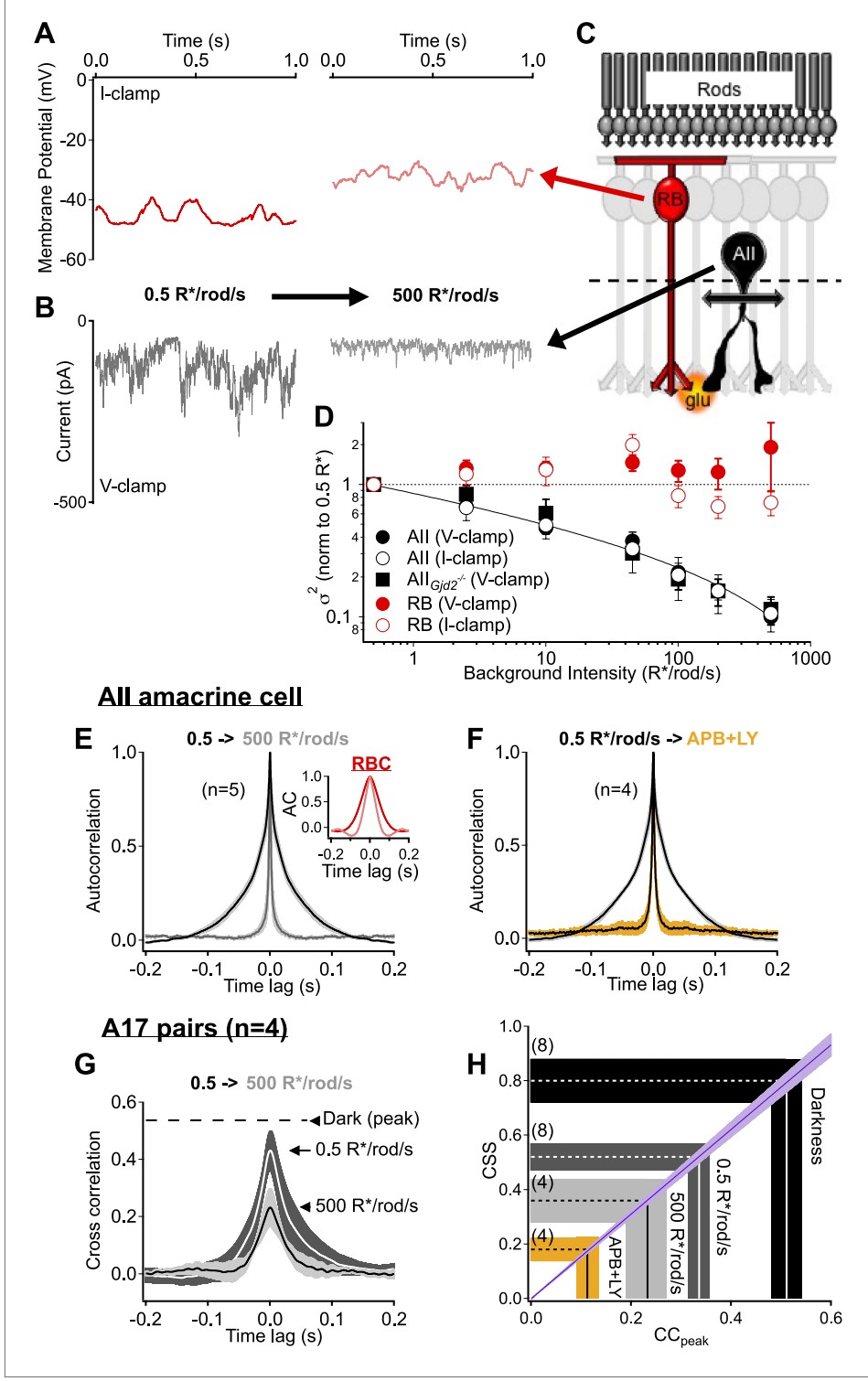

**Figure 6**. Rod bipolar cell output synapses continue to desynchronize with increasing luminance, reducing the transmission of rod-dependent noise at higher backgrounds. (**A–C**) Individual recordings from a RBC (**A** and **C** red) and AII amacrine cell (**B** and **C** *black*) in the presence of a dim background (0.5 R*/rod/s, left) and 1000-fold brighter background (right) illustrate the noise reduction across the RBC. (**D**) Population data for voltage-clamp recordings of excitatory synaptic input and current clamp recordings of membrane signaling in RBCs (*red*) and AII amacrine cells (*black*). Noise recorded from RBCs remained relatively constant across this range of backgrounds (comparison
*Figure 6. Continued on next page*

*Figure 6. Continued*

of noise at 500 relative to 0.5 R*/rod/s, V-clamp: p = 0.92, n = 7; I-clamp: p = 0.18, n = 4) while noise recorded from AII amacrines was reduced ~10-fold (comparison of noise at 500 relative to 0.5 R*/rod/s, V-clamp: p = 0.0039, n = 5; I-clamp: p = 0.019, n = 5). AII amacrine cell recordings from the retinas of mice lacking connexin36-containing gap junctions (*black* squares) indicate that neither gap junctions, nor the secondary rod pathway, are required for this transition (comparison of noise at 500 relative to 0.5 R*/rod/s, V-clamp: p = 0.017, n = 5). Error bars represent ± SEM across cells. (**E**) Average autocorrelation functions for a population of AII amacrine cells recordings under steady-state illumination at 0.5 and 500 R*/rod/s. Slower temporal correlations in the input currents are strongly reduced in the AII amacrine cell across this range of luminance. *Inset*: the reduction in temporal correlations of the RBC voltage response is much less than that observed in the AII. (**F**) The 'LY+APB' manipulation greatly reduces temporal correlations in RBC output, similarly to adaptation to 500 R*/rod/s. (**G**–**H**) Paired-recordings from A17 amacrine cells reveal that RBC output synapses become increasingly desynchronized/independent as the retina is adapted to higher luminance (CC$_{peak}$ = 0.23 ± 0.05 at 500 R*/rod/s vs 0.54 ± 0.06 in darkness, n = 4 pairs, p = 0.029). Taken together these data indicate that RBC synapses desynchronize and reduce rod-dependent noise transmission when the retina is adapted to brighter conditions, when the cone-driven circuits are beginning to convey more of the visual information. Thick lines represent means, shaded regions represent ±SEM.

(***Figure 7D***). Adding the mGluR6 antagonist LY341495 to restore RBC release to near-dark levels (but with low CSS, see ***Figure 5***) produced weak correlations in synaptic input (CC$_{peak}$ = 0.06 ± 0.05, n = 4 pairs).

Thus, correlations in the RGC synaptic inputs and CSS observed in the RBC synaptic output share a similar dependence on normal dendritic input to RBCs from rod photoreceptors, suggesting that high CSS at the RBC→AII synapse plays an important role in creating ganglion cell correlations in darkness. A combination of CSS and redundant connections between RBCs and AII amacrine cells help to amplify and transmit small modulations in RBC voltage, whether noise or signals, and by doing so mitigate the impact of noise intrinsic to downstream synapses.

## Discussion

By exploiting the characteristic connectivity between RBCs and two downstream targets, we investigated how cross-synaptic synchrony influences the transmission of physiological inputs through the retina. We found that synchrony across different RBC output synapses was very strong in the dark, such that ~80% of the vesicle release events occurring at different ribbons within the same RBC axon terminal were synchronous. The high level of cross-synaptic synchrony caused noise in the synaptic inputs to AII amacrine cells to be dominated by noise generated in the rod photoreceptors rather than noise generated at the RBC output synapse itself. As light level increased, conditions under which other retinal circuits become active (***Manookin et al., 2008***), vesicle release at the RBC→AII synapse transitioned from high CSS to low CSS, thus limiting transmission of upstream signals and noise. We discuss these conclusions and their implications for retinal and neural processing in more detail below.

### Implications for photon detection

The ability of the dark-adapted visual system to detect a small number of absorbed photons (***Hecht et al., 1942***) places considerable constraints on the underlying mechanisms. For example, behavioral sensitivity requires that rod photoreceptors detect single photons, and we now have a good understanding of how that is achieved mechanistically (reviewed by ***Rieke and Baylor, 1998***). Behavioral sensitivity also requires that retinal synapses maintain low noise so as not to obscure the single photon responses of the rods. Similar requirements on the fidelity of synaptic transmission arise in other neural circuits that sense subtle changes in input.

Synaptic noise first threatens the fidelity of visual signaling at the synapse between rods and RBCs. This is an unusual sign-inverting synapse in which ongoing release of glutamate from the rod photoreceptors acts via metabotropic glutamate receptors to close ion channels in the RBC dendrites (***Slaughter and Miller, 1981***). Mean release rate at these synapses is highest in the dark, with light exposure leading to a decrease in release rate. In complete darkness, saturation of the postsynaptic metabotropic cascade suppresses the transmission of rod noise; this synaptic saturation enhances the sensitivity of retinal signals more than 10-fold (***van Rossum and Smith, 1998***; ***Field and Rieke, 2002***; ***Berntson et al., 2004***; ***Sampath and Rieke, 2004***).

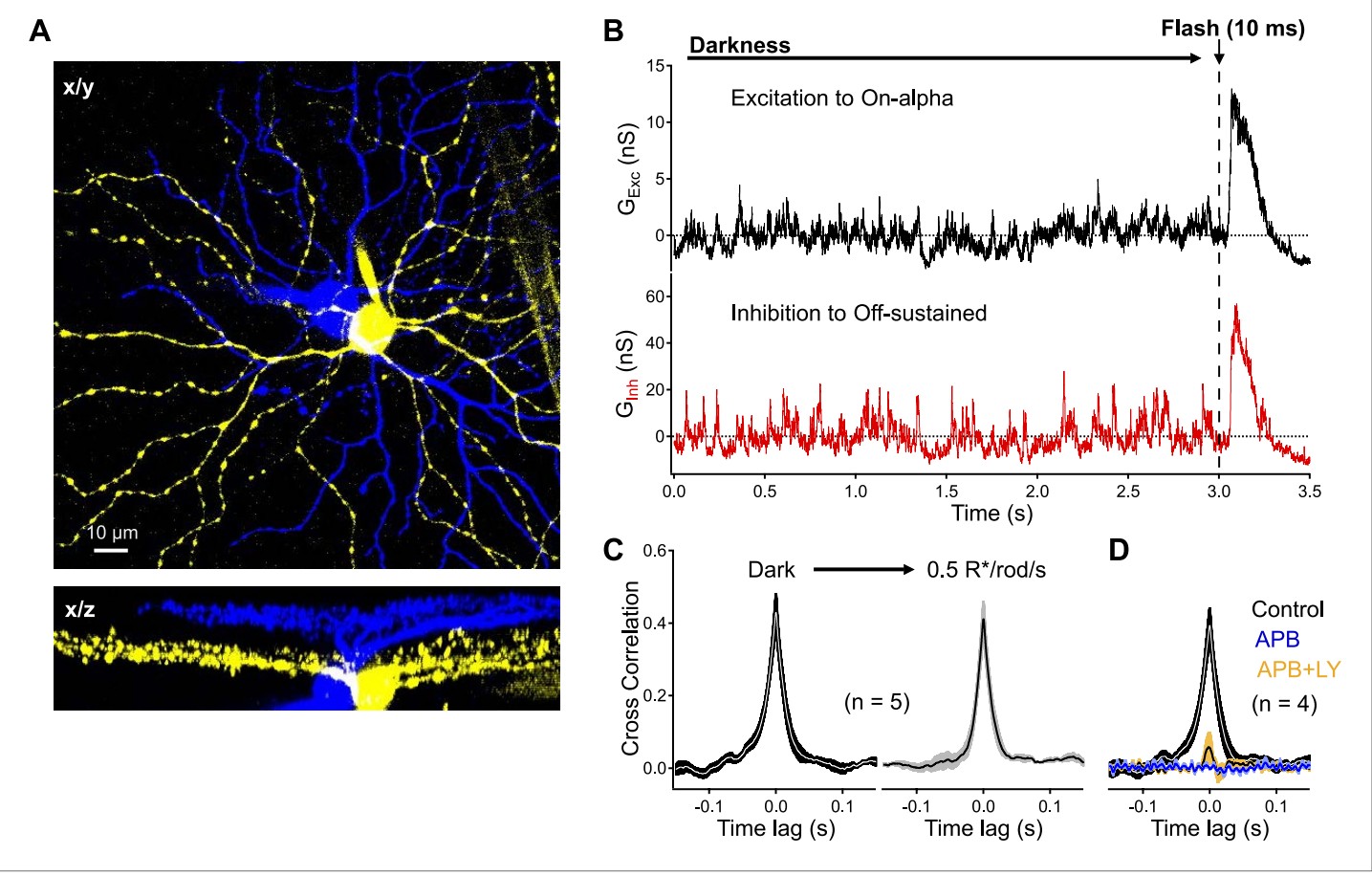

**Figure 7**. Dark noise and CSS drive strongly-correlated synaptic activity in highly-overlapping On alpha - OFF sustained ganglion cell pairs. (**A**) Confocal reconstruction of a paired recording from an On alpha (*yellow*) and OFF sustained (*blue*) ganglion cell. (**B**) Example traces of simultaneous recordings of excitatory input to an On alpha (*black*) and inhibitory input to an Off sustained (*red*) ganglion cells. At the end of each recording epoch a brief flash was delivered to monitor sensitivity over time. (**C**) Peak cross correlations measurements from five pairs indicate that synaptic activity is slightly more correlated in the darkness than in the presence of a dim constant background (p = 0.037 for a change in $CC_{peak}$ relative to dark, n = 5 pairs). (**D**) Suppression of outer retinal activity transmission with APB (5–10 μM) eliminates ganglion cell correlations (p = 0.0048 for change in $CC_{peak}$ relative to dark, n = 4 pairs). Additional application of LY (at similar concentrations to *Figure 5*) produces weak correlations in the pairs (p = 0.0089 for a change in $CC_{peak}$ relative to dark, n = 4 pairs). Thick lines represent means, shaded regions represent ±SEMs. These experiments were conducted using wild type whole mount retinal preparations.

The potential pitfalls associated with synaptic noise recur as signals traverse the retina, mainly via conventional chemical synapses. Our work here details how the operation of synapses between RBCs and AII amacrine cells maximizes sensitivity to upstream activity while minimizing added noise in darkness. Specifically, highly synchronized vesicle release across parallel RBC output synapses amplifies RBC voltage fluctuations produced by rod signals and noise while minimizing added synaptic noise. Quite surprisingly, achieving the estimated dark CSS of 0.8 requires that the dark fluctuations in RBC voltage produce near-deterministic changes in vesicle release at individual synapses (*Figure 8A*). Multiple release sites at each ribbon synapse (each RBC ribbon has ~10 active zones; *Figure 8B*), high vesicle availability and postsynaptic receptor saturation (*Tong and Jahr, 1994*) (*Figure 8C*) will facilitate low synaptic variability. The low rate of ongoing release in darkness compared to that in the presence of steady light ensures that the readily releasable vesicle pool in the RBC terminal remains full (*Singer and Diamond, 2006*), an important component of strong synaptic synchronization. These results highlight that the RBC output synapses operate far from a regime in which vesicle release follows Poisson statistics, and this is important for their ability to reliably transmit the small signals forming the basis of night vision.

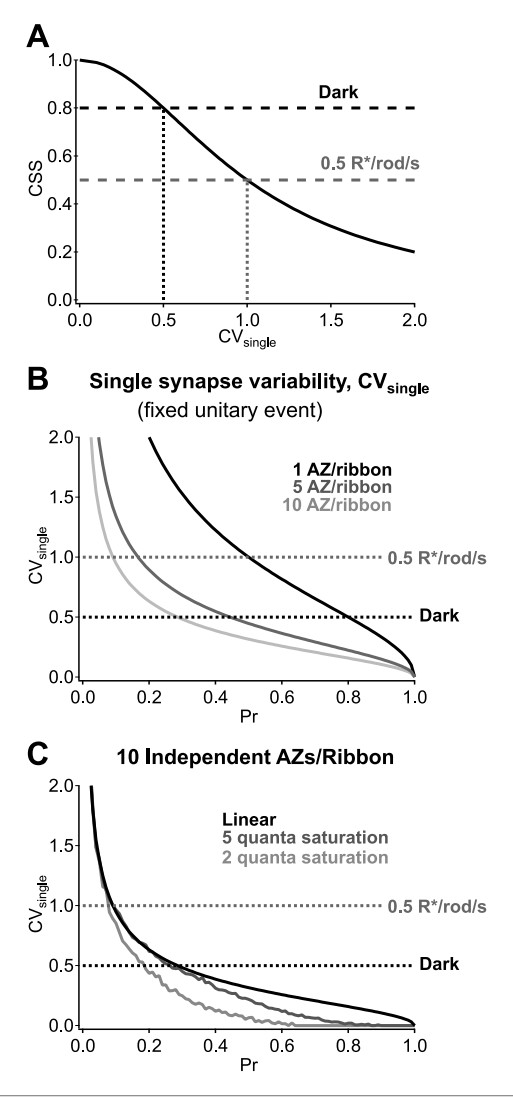

**Figure 8**. Multiple active zones (AZ) per synapse and low synaptic variability enhance CSS in darkness. (**A**) CSS measurements constrain synaptic variability at individual synaptic connections. Assuming a homogeneous release probability, a CSS measurement of 0.8 in darkness indicates that the coefficient of variation (CV) at individual synapses must be ≤0.5. (**B**) Multiple release sites/active zones improve reliability at individual synapses. Previous work indicates that each RBC ribbon synapse has ~10 active zones, thus facilitating multivesicular release. (**C**) Postsynaptic receptor saturation can further improve reliability at individual synapses. If the synaptic receptors are saturated by the release of five or more vesicles (dark gray) then the response to the release of >5 vesicles will be identical to the response to five vesicles. This reflects a tradeoff between dynamic range and reliability.

The control of synaptic output via fluctuations in RBC voltage requires fine control of the synaptic operating point. As described above, if the RBC is too depolarized in the dark the spontaneous release rate will go up (*Jarsky et al., 2011*), which will both increase stochastic fluctuations in release and deplete the pool of releasable vesicles. If the RBC is too strongly hyperpolarized, the synapse will be ineffective in transmitting single photon responses generated in the rods. Thus effective transmission of single photon responses requires a balance, maintained through the ongoing level of dendritic input to the RBC from the rods and inhibitory feedback onto the RBC synaptic terminal from A17 amacrine cells.

## Implications for dynamic interactions between parallel circuits

Functional and anatomical work shows that most neural circuits contain multiple cell types organized to process circuit inputs in parallel. This architectural similarity highlights several common motifs in neural computation: (1) divergence via multiple output synapses can produce correlated activity in downstream circuit elements (*Kazama and Wilson, 2009*); (2) integration and processing of functionally dissimilar inputs from different parallel circuits controls computation in many neurons (*Olsen et al., 2007*; *Fischer et al., 2008*; *Schnell et al., 2010*); and (3) neurons and synapses often participate in multiple functional circuits (*Munch et al., 2009*; *Grimes et al., 2014*), a possible outcome of the evolutionary pressures to maximize computational capacity while economizing the necessary biological hardware. Cross-synaptic synchrony impacts each of these issues.

Retinal ganglion cells, like output neurons in many neural circuits, exhibit strong noise correlations (*Arnett and Spraker, 1981*; *Mastronarde, 1983*; *Murphy and Rieke, 2008*; *Cafaro and Rieke, 2010*; *Volgyi et al., 2013*), and these correlations can be dominated by divergent noise from common upstream circuit elements (*Brivanlou et al., 1998*; *Trong and Rieke, 2008*; *Ala-Laurila et al., 2011*). Cross-synaptic synchrony is a key determinant of whether anatomical divergence will produce correlated signals in downstream neurons. When cross-synaptic synchrony is high, postsynaptic targets of a given neuron will receive highly correlated input. Changes in CSS, such as with mean light level as observed here, will then control the strength

of noise correlations produced by divergence, without morphological changes in the circuit connections.

Changes in CSS could also play an important role in the integration of signals from distinct parallel circuits. For example, as light levels increase from darkness, the need for high gain imposed by detecting sparse photons is reduced and visual perception depends less critically on signals traversing the rod bipolar pathway. Over a substantial range of light levels, visual signals elicit simultaneous activity in rod and cone photoreceptors and their associated circuitry (*Naarendorp et al., 2010*). These parallel signals are combined through several shared circuit elements (e.g., AII amacrine cells) before they are transmitted to targets in the central nervous system. Efficient transmission of noisy rod signals through the rod bipolar pathway could jeopardize cone signals under these conditions. As luminance increases the RBC depolarizes, eventually evoking synaptic depression by way of vesicle depletion and $Ca_v$ inactivation (*Singer and Diamond, 2006*; *Jarsky et al., 2011*; *Oesch and Diamond, 2011*), mechanisms that likely underlie the decrease in cross-synaptic synchrony we observe here. The observed decrease in synchrony within the rod bipolar pathway serves to decrease transmission of rod noise to shared downstream circuit elements. Other highly interconnected brain regions might use similar mechanisms for dynamically regulating signal transmission in parallel circuits prior to signal integration.

## Materials and methods

### Electrophysiology

Experiments were conducted on whole mount and slice (200 µm thick) preparations taken from dark-adapted *Gjd2* knockout (*Deans et al., 2001*, *2002*) or wild-type C57/BL6 mice. Retinas were isolated under infrared visualization and stored in oxygenated (95% $O_2$/5% $CO_2$) Ames medium (Sigma, St. Louis, MO) at ~32°C to 34°C. Once under the microscope, tissue preparations were perfused by the same Ames solution at a rate of ~8 ml/min. Isolated retinas were either flattened onto polyL-lysine slides (whole mount) or embedded in agarose and sliced as previously described (*Dunn et al., 2006*; *Murphy and Rieke, 2006*). Retinal neurons were visualized and targeted for whole-cell recordings using video DIC with an infrared light source (>950 nm). Data in *Figures 1–3,5,6* and *Figure 5—figure supplement 1* were collected from retinal slices, whereas data in *Figures 4,7* and *Figure 4—figure supplement 1* were collected from whole mount preparations. To ensure that retinal recordings consistently reflected a dark-adapted state, only one recording (single cell or paired) came from each dark-adapted retina preparation (i.e., slice or whole-mount).

Voltage clamp recordings were obtained using pipettes (RGCs: 2–3 MΩ, AII and A17 amacrine cells: 5–6 MΩ, bipolar cells: 10–14 MΩ) filled with an intracellular solution containing (in mM): 105 Cs methanesulfonate, 10 TEA-Cl, 20 HEPES, 10 EGTA, 2 QX-314, 5 Mg-ATP, 0.5 Tris-GTP, and 0.1 Alexa (488, 555, or 750) hydrazide (~280 mOsm; pH ~7.3 with CsOH). Current clamp recordings used an intracellular solution containing (in mM): 123 K-aspartate, 10 KCl, 10 HEPES, 1 $MgCl_2$, 1 $CaCl_2$, 2 EGTA, 4 Mg-ATP, 0.5 Tris-GTP, and 0.1 Alexa (488, 555 or 750) hydrazide (~280 mOsm; pH ~7.2 with KOH). NBQX (10 µM; Tocris, Briston, United Kingdom), APB (5–10 µM; Tocris), LY341495 (~0.5–2 µM; Tocris), Mibefridil (10 µM), or an inhibitory cocktail (20 µM SR95531, 50 µM TPMPA and 2 µM strychnine; Tocris) was added to the perfusion solution as indicated in *Figures 1,5–7* and *Figure 5—figure supplement 1*. To isolate excitatory or inhibitory synaptic input, cells were held at the estimated reversal potential for inhibitory or excitatory input of ~−60 mV and ~+10 mV. Absolute voltage values were not corrected for liquid junction potentials ($K^+$-based = −10.8 mV; $Cs^+$-based = −8.5 mV).

### Visual stimuli

For all experiments, full field illumination (diameter: 560 µm) was delivered to the preparation through a customized condenser from blue (peak power at 470 nm) or green (peak power at 510 nm) LEDs.

### Analysis

We estimated the strength of CSS in the RBC output based on paired recordings from highly overlapping A17 amacrine cells and the calculation outlined below. The calculation sums over circular disks centered on the soma. The variance in the response of a single cell is then

$$\sigma^2_{total} = \sum_r n_r \exp(-2(r + r_0)/\gamma)$$

(2)

where $n_r$ is the number of synaptic contacts and $\gamma$ is the electrotonic scaling factor for synaptic inputs at a particular radial distance (the electrotonic length factor for A17 dendrites comes from *Grimes et al., 2010*). A17 dendrites traverse ~40 µm of the inner plexiform before reaching sublaminas 4

and 5 (where they make the majority of their synaptic contacts with RBCs); $r_0$ accounts for the length of these initial descending dendrites. The common variance can then be defined as

$$\sigma_{shared}^2 = \beta_{sync} \sum_r P_{shared}(r) n_r \exp(-2(r + r_0)/\gamma)$$

(3)

where $\beta_{sync}$ is the strength of CSS of RBC synapses (between 0 and 1), and $P_{shared}$ is the percentage of contacted RBCs that are common to the two cells within a particular ring. The cross correlation function at zero time lag is mathematically defined as the shared variance over the geometric mean of the independent variances; in terms of CSS and the A17 paired connectivity the cross correlation function can be defined as in *Equation 1*.

TaroTools event detection plug-ins (for Igor Pro) were used to examine postsynaptic currents in darkness from AII amacrine cells from *Gjd2* knockout mice. Miniature excitatory postsynaptic currents were identified by setting the event detection threshold to −2 pA and requiring a 10–90% rise time of 1 ms or less. Fluctuating baselines (likely due to gap junction input) prohibited effective event detection in WT AIIs.

All data are presented as mean ± SEM and two-tailed paired student's $t$ tests were used to test significance unless otherwise noted.

## Synaptic modeling

With the assumption that synapses in the RBC's axon exhibit homogeneous release probability, the cross synaptic synchrony can be related to the variability at individual synapses as

$$\beta_{sync} = \frac{1}{1 + CV_{single}^2}$$

(4)

where $CV_{single}$ is the coefficient of variation for transmission at a single synapse. Variability at individual synapses was modeled explicitly in *Figure 8* by examining probabilistic signaling for a varying number of independent release sites/active zones at a given ribbon synapse (with fixed vesicle availability and quantal response). Variability in transmission was calculated assuming no variability in the postsynaptic response to a quantal release event as

$$CV_{single} = \frac{\sqrt{1 - Pr}}{\sqrt{N * Pr}}$$

(5)

where $N$ is the number of release sites/active zones at a single ribbon synapse. In cases when two or more vesicles are synchronously released at a single synapse, postsynaptic receptors could experience saturation (*Tong and Jahr, 1994*). This possibility was modeled by equating postsynaptic responses to 2 or more released vesicles. 200 trials were run for each value of Pr with 10 active zones at a given ribbon. For each trial, a release event was initiated when a randomly generated number was less than Pr. The coefficient of variation was then calculated across trials (SD/mean).

## Cell identification

On alpha-like RGCs, Off sustained RGCs, RBCs, A17 amacrine cells, and AII amacrine cells were identified by soma morphology and electrophysiological characteristics. Cell identity was often further confirmed post-recording by imaging the dye-filled arbors (Alexa 488, 555 or 750) using confocal microscopy or epifluorescence.

## Light adaptation

For experiments probing different levels of luminance (*Figures 2,6 and 7*), we allowed 30–120 s of adaptation at each luminance level before steady state was reached, and data were analyzed.

## Cell selection criteria

Recordings from slice preparations were performed within ~4 hr of retinal dissection, and we specifically targeted neurons that were ≥20 μm below the surface of the slice. RBCs were selected for when saturating flashes from darkness produced reliable and robust events both before and after light adaptation. RBC recordings were kept short (typically 2–5 min) to minimize washout effects. AII amacrine cells could be targeted particularly deep in the slice (~40–50 μm) and provided stable long lasting recordings (~30 min). To maximize the overlap in RBC sampling, we targeted pairs of A17 amacrine cells whose cell bodies were separated by less than 80 μm.

## Assessment of A17→RBC connectivity

To target A17 cells in whole mount mouse retina, the BAC Igfbp2 Gensat transgenic line (www.gensat. org) was reconstituted from cryo-frozen sperm (FVB background, stock# 030560-UCD, www.mmrrc. org) using in vitro fertilization of Cd1/C57 hybrid eggs; mice were then bred into a C57/BL6 background. We confirmed morphologically that amacrine cells labeled in this line were A17 amacrines (*Siegert et al., 2009*) (*Figure 4—figure supplement 1*). GFP-positive A17 amacrine cells were targeted in 3- to 6-week-old mice. A17 cells were injected using sharp electrodes (tip resistance ~150 MΩ) with 2% Lucifer yellow (in 200 mM KCl) prior to fixation with 4% paraformaldehyde in 0.1 M phosphate buffered saline (PBS) for 20 min. The retinas were rinsed with PBS and incubated for 72 hr with rabbit polyclonal lucifer yellow (1:500, Invitrogen, Carlsbad, CA) and mouse monoclonal PKC (1:500, Sigma) antibodies in PBS with 0.5% Triton and 5% donkey serum. Retinas were incubated with secondary antibodies (anti-rabbit Alexa Fluor conjugate, Invitrogen and anti-mouse DyLight conjugate, Jackson ImmunoResearch, West Grove, PA) for 12 hr in PBS. Retinas were then mounted in Vectashield (Vector labs, Burlington, CA).

Images were acquired with an Olympus FV1000 microscope using a 1.35 NA 60× oil objective, at a voxel size of 0.102 × 0.102 × 0.3 μm or 0.204 × 0.204 × 0.3 μm. Raw image stacks were processed with MetaMorph (Universal Imaging) and Amira (Mercury Computer Systems). To identify sites of apposition between PKC positive RBCs and A17 amacrine cell varicosities, pixel overlap was assessed upon rotation of the image volumes in 3D using Amira. Synaptic contact was defined when the fluorescent signals overlapped by >1 pixel at all angles of the 3D rotation. To determine the percentage of RBCs in the field of view that contacted the A17 cell, individual RBCs were digitally isolated and reconstructed using the 'label-field' function of Amira. To generate connectivity maps of the A17s, RBCs axonal locations were assessed in concentric rings spaced 20 μm apart, centered on the soma of the injected A17. A17 dendrites traverse ~40 μm of the inner plexiform before reaching sublaminas 4 and 5 (where they make the majority of their synaptic contacts with RBCs); therefore, the most central concentric ring corresponds to dendritic distances between 40 and 60 μm, the second ring corresponds to dendritic distances between 60 and 80 μm and so on.

The A17→RBC connectivity patterns derived from whole mount preparations were used to derive the slope factor that relates the RB's CSS to correlations measured in highly overlapping A17s (*Equation 1* and *Figure 4G*). To account for the effects of slicing in our calculations, we divided the number of contacted RBCs within a given ring by two (except for the first ring). Using these values and *Equation 1*, we estimate the slope (purple line in *Figure 4G*) to be 0.64 ± 0.03 for highly overlapping A17s recorded in the slice preparation (0.67 ± 0.03 for whole mount).

## Electron microscropy

A previously published data set was analyzed (*Briggman et al., 2011*) (retina k0563). Voxel dimensions were 12 × 12 × 25 nm$^3$. Segmentation of identified RBCs, AIIs, and A17s were performed using ITK-SNAP (*Yushkevich et al., 2006*) (www.itksnap.org) and rendered in Matlab.

## Acknowledgements

We thank Sid Kuo, Gautam Awatramani, Gabe Murphy for helpful comments on earlier versions of the paper, Felice Dunn for assistance with preliminary data collection and Mike Ahlquist, Mark Cafaro, Shellee Cunnington, and Paul Newman for outstanding technical assistance. Support provided by HHMI (FR), NIH Extramural Funding (EY11850 to FR, EY10699 to ROW) and the NINDS Intramural Research Program (KLB).

## Additional information

### Funding

| Funder | Grant reference number | Author |
| --- | --- | --- |
| National Eye Institute | EY11850 | Fred Rieke |
| Howard Hughes Medical Institute | | Fred Rieke |
| National Eye Institute | 10699 | Rachel O Wong |

The funders had no role in study design, data collection and interpretation, or the decision to submit the work for publication.

## Author contributions

WNG, MH, Conception and design, Acquisition of data, Analysis and interpretation of data, Drafting or revising the article; KLB, Acquisition of data, Analysis and interpretation of data, Drafting or revising the article; ROW, FR, Conception and design, Analysis and interpretation of data, Drafting or revising the article

## Ethics

Animal experimentation: This work was performed in strict accordance with the recommendations in the Guide for the Care and Use of Laboratory Animals of the National Institutes of Health. All procedures followed protocols approved by the Institutional Animal Care and Use Committee (protocol 3030-01) of the University of Washington.

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
