## [Decision Letter]

Thank you for sending your work entitled “Cross-synaptic synchrony and transmission of signal and noise across the mammalian retina” for consideration at *eLife.* Your article has been favorably evaluated by Eve Marder (Senior editor) and 3 reviewers, one of whom is a member of our Board of Reviewing Editors.

The following individuals responsible for the peer review of your submission have agreed to reveal their identity: Ronald L Calabrese (BRE/reviewer #1) and Noga Vardi (reviewer #2). Reviewer #3 remains anonymous.

The Reviewing editor and the other reviewers discussed their comments before we reached this decision, and the Reviewing editor has assembled the following comments to help you prepare a revised submission.

This work of Grimes at al contrasts synaptic noise in AII vs A17 amacrine cells of the mouse, exploring in particular the correlations in synaptic release among multiple release sites from retinal bipolar cells onto amacrine cells. The main experimental design takes advantage of an anatomical observation that each A17 amacrine cell receives only one synaptic input from a particular RBC. This allowed the authors to measure the correlation between two neighboring A17s, and use it as an estimate of the correlation that results from two different synapses located in the same RBC. They then went on and calculated the CSS and evaluated correlation in the next cell order (RGCs). The conclusions are quite striking. In very low light levels different synaptic sites on the same retinal bipolar cell show highly correlated releases, almost deterministic/synchronous. Presumably this arises from voltage fluctuations in the terminals that are shared amongst release sites and have strong effects on release probability. As a result, noise at individual synaptic sites is diminished in favor of a concerted noise generated in the bipolar cell as a result of its responses to rod input. An interesting downstream result of this is that releases of glutamate and of glycine onto ganglion cells also show a shared pattern of noise. As light levels increase, such correlations, i.e., sensitivity to bipolar cells voltage fluctuation, decreases.

The reviewers were consistent in their praise of the strong experimental design and the elegant experiments; their only concerns were with the presentation. The authors can satisfy these concerns by some careful rewriting and more discussion. Reviewer #1 would like the authors to make it clearer that what they really want to show is CSS at the RBC to AII synapses and that the RBC to A17 synapses are used as a tool to estimate CSS at the AII synapses. Reviewer #2 praises the paper for putting their aims for the study up front but would like the authors to make Aim 3 conform more to what was actually achieved. Reviewer #2 would also like to see electrical compactness of the RBC addressed in the context of CSS. Please also address the minor comments.

*Reviewer #1 minor comments*:

1) Please identify the experimental system (mouse) in the title to meet *eLife* requirements. For example, the title might be “Cross-synaptic synchrony and transmission of signal and noise across the murine retina.”

2) The Abstract was very confusing to me with its focus on A17 cells and stating that “...A17s often receive input from different synapses made by the same rod bipolar cell...” Do you mean AIIs? Can you explain in the Abstract that AIIs are the focus and A17 the tools without going over the word limits?

*Reviewer #2 minor comments*:

From the Results section:

1) I assume you meant that one-rod bipolar cell releases glutamate at 50 ribbon synapses?

2) I find the sentence in parenthesis confusing because of “i.e.,”; maybe it will be clearer to say that “Both cell types receive … as can be inferred from…”.

3) An important factor that may differentiate the noise levels in AII and A17, which has been ignored in this paper, is the cells’ size. AII is relatively small and may have a higher input resistance.

4) For future reference, 2P imaging can work very nicely in a slice when using region of interest that do not include photoreceptors.

5) Please define CCpeak (since Methods comes after Results).

6) Why is it surprising to have this correlation: I think it is expected given that compactness of the RBC.

---

## [Author Response]

*The reviewers were consistent in their praise of the strong experimental design and the elegant experiments; their only concerns were with the presentation. The authors can satisfy these concerns by some careful rewriting and more discussion. Reviewer #1 would like the authors to make it clearer that what they really want to show is CSS at the RBC to AII synapses and that the RBC to A17 synapses are used as a tool to estimate CSS at the AII synapses*.

This is an excellent point. We now emphasize the distinction in how we use AII and A17 amacrine cells in both the Abstract and the beginning of the Results section. We specify in the Abstract that ‘The anatomical connectivity between rod bipolar and A17 amacrine cells in the mammalian retina…provides a rare technical opportunity to measure cross-synaptic synchrony.’ A few sentences later we emphasize the role of CSS in transmission of signal and noise to downstream cells. We also revised the second paragraph of Results to more explicitly state the differences in how we use the two amacrine cells: ‘AII and A17 amacrine cells provide the two main postsynaptic targets of RBCs. As described below, the connectivity between RBCs and these postsynaptic neurons provide an opportunity to 1) measure cross-synaptic synchrony (CSS) – using paired A17 recordings and 2) examine its role in neural transmission – by recording feedforward signaling in the AII amacrine cell and other downstream circuit elements.’

*Reviewer #2 praises the paper for putting their aims for the study up front but would like the authors to make Aim 3 conform more to what was actually achieved*.

We have revised that sentence to more clearly state that our aim was to determine the role of CSS in transmission of both signal and noise through the rod bipolar circuit. The previous version emphasized signal alone, and indeed much of our work focuses on how CSS controls noise intrinsic to the rod bipolar output synapse and hence permits noise from the rods themselves to be the primary source of noise in responses of downstream neurons.

*Reviewer #2 would also like to see electrical compactness of the RBC addressed in the context of CSS. Please also address the minor comments*.

We now state that ‘Thus strong synaptic synchronization requires large, coordinated increases in the probability of release (a notion supported by the electrically-compact nature of the RBC, [41]) and low intrinsic variability at individual synaptic connections (see Discussion; Figure 8).’

Reviewer #1 minor comments:

*1) Please identify the experimental system (mouse) in the title to meet* eLife *requirements. For example, the title might be “Cross-synaptic synchrony and transmission of signal and noise across the murine retina*.*”*

The title has now been changed to ‘Cross-synaptic synchrony and transmission of signal and noise across the mouse retina’.

*2) The Abstract was very confusing to me with its focus on A17 cells and stating that “...A17s often receive input from different synapses made by the same rod bipolar cell...” Do you mean AIIs? Can you explain in the Abstract that AIIs are the focus and A17 the tools without going over the word limits*?

Thanks for pointing out this confusing description and the need to emphasize A17s use as a tool. See the response to the first general comment above, which details changes in both the Abstract and Results to clarify the differences in how we use A17 recordings and AII recordings.

Reviewer #2 minor comments:

*From the Results section*:

*1) I assume you meant that one-rod bipolar cell releases glutamate at 50 ribbon synapses*?

The sentence now states that ‘Each rod bipolar cell releases glutamate from ∼50 ribbon synapses ([52]; [56]; Singer et al., 2004; [57]; Figure 1).’

*2) I find the sentence in parenthesis confusing because of “i.e.,”; maybe it will be clearer to say that “Both cell types receive … as can be inferred from…”*.

The sentence now states that ‘Both cell types received substantial excitatory input in complete darkness as evinced byµMthe suppression of holding current and noise by 10 NBQX, an AMPA receptor antagonist (data not shown); however, spontaneous current fluctuations…’

*3) An important factor that may differentiate the noise levels in AII and A17, which has been ignored in this paper, is the cells’ size. AII is relatively small and may have a higher input resistance*.

We now mention the possibility of differences in electrical properties: (‘Multiple factors, e.g., differences in the cells’ electrical properties…’). Rather than explore this possibilities, however, we favor a quick transition into the A17 paired recordings which provide direct insights into synaptic synchronization, which is the main focus of the paper.

*4) For future reference, 2P imaging can work very nicely in a slice when using region of interest that do not include photoreceptors*.

We’ve slightly modified that sentence to say ‘The RBC’s CSS cannot be measured under dark-adapted conditions using imaging approaches because even two-photon (i.e. infrared) imaging produces too much rod activation to maintain the retina in a dark-adapted state (18).’ Indeed, while we also find that 2P imaging works well in a slice, even in that case there is sufficient light activation of rods to preclude measurements of fully dark-adapted retinal responses.

*5) Please define CCpeak (since Methods comes after Results)*.

We changed this to read ‘Paired recordings from neighboring A17 amacrine cells (distance between somas < 80 ∝m) revealed strong correlations in excitatory synaptic input in the dark (The peak of the cross-correlation function in darkness, i.e. the Dark CC_peak_, was 0.51±0.02, n = 8 pairs; Figure 2).’

*6) Why is it surprising to have this correlation: I think it is expected given that compactness of the RBC*.

We found the strength of correlations surprising given the lack of visual stimuli (in the dark-adapted prep) and the lack of spiking behavior in RBCs. We’ve now modified the paragraph to explicitly mention these issues and to explicitly mention the electrically compact nature of the RBCs. Specifically, we now say: ‘The strength of correlations in spontaneous inputs to neighboring A17 amacrine cells in darkness indicated a surprising level of synaptic synchronization considering the lack of visual stimuli in these conditions and the fact that the presynaptic RBCs are non-spiking neurons. Thus even if the two recorded A17s receive input from an identical set of RBCs, correlation strengths near 0.5 require that two synapses made by the same RBC must be coactive at least half the time. Thus strong synaptic synchronization requires large, coordinated increases in the probability of release (a notion supported by the electrically-compact nature of the RBC, [41]) and low intrinsic variability at individual synaptic connections (see Discussion; Figure 8).’

We also delve into this issue more completely in Figure 8, illustrating the large changes in the probability of release that are required to explain the measured CSS strength.